



# Age and origin of leaf wax *n*-alkanes in fluvial sediment-paleosol sequences, and implications for paleoenvironmental reconstructions

Marcel Bliedtner[1,2], Hans von Suchodoletz[3,4], Imke Schäfer[2], Caroline Welte[5], Gary Salazar[6], Sönke Szidat[6], Mischa Haas[7], Nathalie Dubois[8], Roland Zech[1]

[1]Institute of Geography, University of Jena, Löbdergraben 32, 07743 Jena, Germany
[2]Institute of Geography and Oeschger Centre for Climate Change Research, University of Bern, 3012 Bern, Switzerland
[3]Institute of Geography, University of Technology Dresden, Dresden, Germany
[4]Institute of Geography, University of Leipzig, 04103 Leipzig, Germany
[5]Laboratory of Ion Beam Physics, ETH Zurich, 8093 Zurich, Switzerland
[6]Department of Chemistry and Biochemistry and Oeschger Centre for Climate Change Research, University of Bern, 3012 Bern, Switzerland
[7]Department of Surface Waters Research and Management, Eawag, 8600 Dübendorf, Switzerland
[8]Department of Earth Sciences, ETH Zürich, 8006 Zürich, Switzerland

*Correspondence to:* Marcel Bliedtner (marcel.bliedtner@uni-jena.de)

**Abstract.** Leaf wax *n*-alkanes are increasingly used for quantitative reconstructions of past environmental changes. However, this is complicated in sediment archives with associated hydrological catchments since the stored *n*-alkanes can have different ages and origins. Direct $^{14}$C-dating of the *n*-alkanes has a great potential to derive independent age information for these proxies, allowing their correct paleoenvironmental interpretation. This holds also true for fluvial sediment-paleosols sequences (FSPS), where the *n*-alkane signal is divided into (i) a catchment signal that is deposited with fluvial sediments, and (ii) an *in-situ* signal from local biomass that increasingly dominates (paleo)soils with time. Therefore, age and origin of *n*-alkanes in FSPS are complex: In fluvial sediment layers they can be pre-aged and reworked when originating from eroded catchment soils, or originate from organic-rich sediment rocks in the catchment. In (paleo)soils, besides an inherited contribution from the catchment they were formed *in-situ* during pedogenesis and originate from local biomass. Depending on the different relative contributions from these sources, the *n*-alkane signal from an FSPS can show variable age offsets between its formation and final deposition.

During this study, we applied compound-class $^{14}$C-dating to *n*-alkanes from a fluvial sediment-paleosol sequence (FSPS) along the upper Alazani River in eastern Georgia with Jurassic black clay shales in its upper catchment. Our results show that pre-heating the *n*-alkanes with 120°C for 8h before $^{14}$C-dating effectively removed the shorter chains (<$C_{25}$), including a part of the catchment-derived *n*-alkanes from Jurassic black clay shales. The remaining petrogenic contributions on the longer chains (≥$C_{25}$) were estimated and corrected for using a constant correction factor. This was based on *n*-alkane concentrations in a Jurassic black clay shale sample from the upper catchment. Due to different degrees of pre-aging and reworking, the corrected leaf wax *n*-alkane ages still indicate relatively large age offsets between *n*-alkane formation and deposition: While due to a dominance of *in-situ* produced leaf wax *n*-alkanes no age offsets existed in intensively developed (paleo)soils, much larger age offsets in less intensively developed paleosols indicate larger proportions of inherited leaf wax *n*-alkanes from the fluvial



parent material. Accordingly, age offsets in non-pedogenic fluvial sediments were largest, and strongly increase after ~4 ka cal. BP due to a greater proportion of pre-aged and reworked leaf wax *n*-alkanes. The leaf wax *n*-alkanes from intensively developed (paleo)soils indicate a local dominance of grasses/herbs throughout the Holocene, which was most likely caused by anthropogenic activity. The leaf wax *n*-alkanes from fluvial sediments show a dominance of both deciduous trees/shrubs and

grasses/herbs in different parts of the FSPS between ~8 and ~5.6 cal. ka BP. Since no older deciduous trees/shrub-derived *n*-alkanes from the catchment were dated, this seems to confirm the formerly proposed delayed regional post-glacial reforestation compared with western and central Europe.

# 1 Introduction

Long-chain *n*-alkanes ($\geq C_{25}$) are valuable biomarkers that are biosynthesized as epicuticular leaf waxes by higher terrestrial

plants. They stay well preserved in soils and sediment archives because of their low water-solubility, chemical inertness and persistence against degradation (Eglinton and Eglinton, 2008). The homologue distribution of leaf wax *n*-alkanes, as well as their stable hydrogen and carbon isotopic composition, are increasingly used to quantitatively reconstruct past changes in vegetation and hydro-climatic conditions from various sediment archives, including lake sediments (Sauer et al., 2001; Schwark et al., 2002; Sachse et al., 2006; Wirth and Sessions, 2016), loess-paleosol sequences (Schäfer et al., 2018; Häggi et

al., 2019), marine sediments (Schefuß et al., 2005) and fluvial sediment sequences (Bliedtner et al., 2018a). However, reconstructions from sediment archives with associated hydrological catchments can be complicated by the fact that sediments and leaf wax *n*-alkanes transit through the catchment over hundreds to several thousands of years, and thus the timing of their deposition does not necessarily reflect the timing of leaf wax *n*-alkane formation (Smittenberg et al., 2006; Feng et al., 2013; Douglas et al., 2014; Gierga et al., 2016; Douglas et al., 2018). Resulting age offsets between leaf wax *n*-alkane formation and

deposition will therefore limit any quantitative paleoenvironmental reconstruction.

In general, organic carbon (OC) and *n*-alkanes that are preserved in fluvial, lacustrine and marine sediment archives can originate from different sources (Hedges et al., 1986). They can: (i) directly derive from recent to sub-recent plant biomass, (ii) be pre-aged and reworked when derived from eroded catchment soils that were formed before their erosion and final deposition in the sediment archive (Blair and Aller, 2012), and (iii) originate from organic-rich sediment rocks, and this highly

aged petrogenic organic material has undergone alteration and has to be regarded as fossil, i.e. it is [14]C-dead (Galy et al., 2008; Hilton et al., 2010). These different sources of OC and *n*-alkanes explain the wide range of [14]C-ages that are reported in the literature for riverine transported particulate organic matter (POC) and leaf wax *n*-alkanes and *n*-alkanoic acids (Galy and Eglinton, 2011; Marwick et al., 2015; Tao et al., 2015; Schefuß et al., 2016). Age offsets between leaf wax *n*-alkane formation and deposition in the order of hundreds to thousands of years were mostly reported from lake sediments, and seem to increase

throughout the Holocene due to anthropogenically induced soil erosion (Douglas et al., 2014; Gierga et al., 2016).

Possibilities to investigate the sources of OC and *n*-alkanes and to track their way through hydrological catchments have been made possible by the development of the MIni CArbon-DAting System (MICADAS), that is equipped with a hybrid ion source





and thus enables measurements of samples with very small amounts of carbon (Wacker et al., 2010; Ruff et al., 2011). The MICADAS enables direct [14]C-dating of specific OC compounds, such as specific leaf wax *n*-alkane compounds. However, because compound-specific [14]C-dating requires specialized equipment (preparative gas chromatography) and is very labor- and cost-intensive, compound-class [14]C-dating where the whole *n*-alkane fraction is dated has been proposed as an elegant and

cost-effective alternative at least for loess-paleosol sequences (Haas et al., 2017; Zech et al., 2017).

Up to now, age and origin of leaf wax *n*-alkanes have not been investigated in fluvial sediment-paleosol sequences (FSPS), although such archives can be found ubiquitously in many regions of the world and have a great potential for leaf wax *n*-alkane-based paleoenvironmental reconstructions (Bliedtner et al., 2018a). In such sequences, the fluvial sediments were deposited during phases of geomorphic activity with intensive flooding, and contain *n*-alkanes that carry a mixed catchment

signal. In contrast, the paleosols were formed during phases of geomorphic stability when no or much less fluvial sediments were deposited. Besides catchment-derived *n*-alkanes from previous fluvial sedimentation, they contain *n*-alkanes that derive from local biomass and thus carry an *in-situ* signal (Bliedtner et al., 2018a). Therefore, *n*-alkanes in FSPS have a complex origin, and can either have formed *in-situ* during soil formation, or be pre-aged and reworked or originate from petrogenic sources when derived from the catchment. Depending on different relative contributions from these sources, the *n*-alkane

signal recorded in FSPS can be older than the timing of sedimentation, and the resulting age offsets can vary throughout the profile.

Here we investigated [14]C-ages of *n*-alkanes from a FSPS along the upper Alazani River in eastern Georgia. Besides leaf wax-derived *n*-alkanes, also [14]C-dead petrogenic *n*-alkanes from Jurassic black clay shales from the upper Alazani sub-catchment contribute to all chain-lengths of the *n*-alkane signal in the FSPS ($C_{21}$-$C_{35}$) without a distinct odd-over-even predominance

(OEP) (Bliedtner et al., 2018a). This petrogenic contribution should lead to increased age offsets, and will complicate compound-class [14]C-dating of the *n*-alkanes as a whole fraction. Therefore, to overcome this limitation we removed the mostly petrogenic short-chain *n*-alkanes ($<C_{25}$) in our samples by a pre-heat treatment, and subsequently applied a simple correction approach for the remaining petrogenic contributions that underlie the long-chain *n*-alkanes ($\geq C_{25}$) that are mostly derived from Quaternary leaf waxes. We hypothesized that due to pre-aging effects, i.e. reworking of soil-derived leaf wax *n*-alkanes from

the upper Alazani sub-catchment, the remaining age offsets should be larger for the fluvial sediment layers compared with the well-developed paleosols. Doing so, we wanted to (i) evaluate the potential of *n*-alkane [14]C-dating in FSPS after removing the petrogenic contribution, (ii) disentangle the different *n*-alkane sources and pathways before deposition, and (iii) directly date the *n*-alkane proxies in our investigated FSPS for more robust paleoenvironmental interpretations.

## 2 Material and methods

### 2.1 Studied site

The investigated FSPS (42°02'17.7"N, 45°21'18.7"E; 450 m a.s.l.) is located in the upper Alazani valley in eastern Georgia. The Alazani River originates at an altitude of ~2,800 m a.s.l. from the southern slopes of the Greater Caucasus Mountains (Fig.



1, 2A). The Alazani flows from North to South for the first ~40 km and then, after its confluence with the smaller Ilto River, follows the NW-SE oriented Alazani thrust top basin for ~160 km (Fig. 2A). Here the Alazani is paralleled by the southern foothills of the Greater Caucasus in the NE (Adamia et al., 2010), and the southwesterly advancing Kura-fold-and-thrust-belt (Kura-FTB) in the SW (Forte et al, 2010) (Fig. 1, 2A). Finally, the ~240 km long Alazani drains via the Kura River into the

Caspian Sea (Fig. 1). The investigated FSPS is located in the upper part of the Alazani thrust top basin ca. 10 km downstream of the confluence of Ilto and Alazani River (Fig. 2A). Upstream of this site both rivers show a braided character and are thus characterized by coarse gravelly river beds. The site-related sub-catchment in the upper Alazani and Ilto valleys is mountainous with a relatively small size of ~1,100 km². It shows relatively steep slopes with an averaged slope angle of ~20°, and especially in the uppermost part slopes up to 50° are found (Bliedtner et al., 2018a; Fig. 2A, B). The uppermost part of the upper Alazani

sub-catchment in the central southern Greater Caucasus is formed by folded and metamorphosed Jurassic flysch and molasse deposits that consist of altered organic-rich black clay shales, sand-/siltstones and volcanic rocks (Fig. 2C). Following downstream, the middle part of the upper Alazani sub-catchment is dominated by Cretaceous sand-/silt- and limestones. The southern part of the sub-catchment is dominated by the Kura-FTB that reaches altitudes of 2,000 m, and consists of folded and overthrusted Cretaceous sand-/silt- and limestones, and Paleogene to Quaternary sandstones, siltstones and conglomerates

(Gamkrelidze, 2003; Fig. 2C). Brownish loamy slope deposits are found in parts of the lower Ilto sub-catchment that probably originate from late Pleistocene aeolian loess (Bliedtner et al., 2018a; Fig. 2C).

At the studied site, recent mean annual temperature and precipitation are about 12.0°C and 720 mm/a, respectively (http://de.climate-data.org/location/28480/, station: Akhmeta). In the upper Alazani sub-catchment, precipitation reaches up to 2,000 mm/a because most parts are located in the central southern Greater Caucasus (unpublished precipitation map V.

Bagrationi Geographical Institute Tbilisi). Precipitation mainly falls in spring and early summer during convective events (Lydolph 1977), and thus the Alazani river reaches its maximal discharge between April and June due to both snow melt in the Greater Caucasus and the concomitant precipitation maximum, i.e. shows a pluvio-nival runoff regime (Suchodoletz et al., 2018).

The recent vegetation of eastern Georgia belongs to the Irano–Turanian Group (Connor et al., 2004; Sagheb-Talebi et al.,

2014). The natural vegetation of the floodplains in the upper Alazani valley, where the studied site is located, consists of deciduous elm-oak-vine forests (Connor and Kvavadze, 2008). Today, agricultural fields and grasslands cover most of the recently inactive elevated valley floor that is outcropped by the investigated FSPS, whereas the lower-lying active modern floodplain hosts scattered patches of deciduous riparian forests. The mid-mountain belt further upstream is characterized by mixed beech and in small parts also by fir-spruce forests. Alpine to subalpine meadows cover the highest parts of the catchment

(Connor and Kvavadze, 2008). ~65% of the upper Alazani sub-catchment are covered by forests, and ca. 35% by grasslands and fields today (Bliedtner et al., 2018a).



## 2.2 Stratigraphy of the investigated FSPS

The investigated FSPS is naturally exposed up to ~7 m along the upper Alazani River, and consists mostly of fine-grained overbank sediments with intercalated paleosols. The sequence was formerly investigated by Suchodoletz et al. (2018) where a more detailed description can be found. Throughout the sequence, six blackish-grayish to reddish paleosols were developed in the fine-grained silty to clayey overbank sediments (Fig. 3). Three intensively developed paleosols are characterized by distinct upper but gradual lower limits (Ahb1, Ahb5, Ahb6), whereas three weakly developed paleosols (Ahb2, Ahb3, Ahb4) only showed gradual upper and lower limits. A well-developed recent soil (Ah) covers the surface of the sequence. The sediments from the FSPS were formerly analyzed for carbonate content, total organic carbon (TOC), pH and mass-specific magnetic susceptibility (χ) to differentiate *in-situ* formed paleosols from partly similar-looking fluvial sediments. Based on a relative Soil Development Index (SDI), the differences of these measured values from the uppermost sample of a paleo(soil) to the underlying parent material were averaged to calculate soil development intensities (Suchodoletz et al., 2018). All paleosols are characterized by systematically decreasing carbonate contents and pH-values, but increasing TOC-contents and χ-values from bottom to top.

The chronology of the FSPS is based on nine charcoal pieces that were dated with [14]C (Fig. 3). The [14]C-ages are in stratigraphic order except two samples that are older than stratigraphically lower samples. Therefore, they must overestimate their true burial age, i.e. they might have been reworked. Apart from these two samples, the charcoal chronology of the FSPS represents the timing of sediment deposition/soil development, and serves as an independent age control for comparison with leaf wax *n*-alkane [14]C-ages (Fig. 3). The chronology ranges between 8.0-8.2 cal. ka BP (95.4%) in the upper part of Ahb6, and 1.6-1.8 cal. ka BP (95.4%) in the sediments below the recent soil Ah. The oldest [14]C-age of 8.0-8.2 cal. ka BP (95.4%) was obtained from a charcoal piece that was found together with archeological artefacts (potsherds, bones, obsidian tools) in the upper part of Ahb6. This documents anthropogenic activity at this site at least since the Neolithic period. Based on the [14]C-age of 1.6-1.8 cal. ka BP (95.4%) from the non-pedogenic fluvial sediments below the recent soil (Ah), the time to form the Ah was estimated to ca. 1.6 ka. This age was used as a reference to calculate the approximate soil forming durations for the paleosols based on the Soil Development Index (SDI). For more detailed information how the SDI was derived, the reader is referred to Suchodoletz et al. (2018).

## 2.3 Analytical procedure

### 2.3.1 *n*-Alkane extraction, separation and quantification

24 samples from the FSPS along the upper Alazani River were formerly analyzed over the whole profile for their *n*-alkane homologue distributions (Bliedtner et al., 2018a). 5 of these samples were used for this study as a pre-test. These samples were extracted with accalerated solvent extraction as described by Bliedtner et al. (2018a). Based on this pre-test, 11 samples from the FSPS, including 4 from the pre-test, were extracted again using an ultrasonic treatment according to Bliedtner et al. (2018b).



To guarentee stratigraphical representativeness, 6 of these reextracted samples were choosen from (paleo)soils and 5 from fluvial sediment layers.

All samples for [14]C-dating were extracted from air-dried and sieved (<2 mm) sample material. The total lipid extract of all samples was separated over aminopropyl pipette columns into: i) the apolar fraction including the *n*-alkanes, (ii) the more polar fraction, and (iii) the acid fraction. The *n*-alkanes were eluted with ~4 ml hexane and subsequently purified over coupled silver-nitrate ($AgNO_3$.) – zeolite pipette columns. The subsequent dating approach encompassed two steps:

1. First, the *n*-alkane fractions of 5 samples (Ala 25I, 225I, 290I, 425I, 505I) were tested for compound-class [14]C-dating (i.e. [14]C-dating of the whole *n*-alkane fraction) during the pre-test (see Table 1, samples with BE Nr's.): Whereas the *n*-alkane fractions of four samples were not heated prior to [14]C-measurement, the *n*-alkane fraction of sample Ala 25 I was pre-heated with 120°C for 8h. The pre-heating of the *n*-alkane fraction was carried out in a 1.5 ml GC vial with solvent evaporated and the GC vial placed into an oven at 120°C for 8h. The heated *n*-alkane fraction yielded a much younger [14]C-age compared with the other samples, since the short-chain *n*-alkanes containing a significant petrogenic *n*-alkane contribution ($<C_{25}$) were effectively removed (see Table 1 and Results and Discussion section).

2. Based on the results of the pre-test, the *n*-alkane fractions of the 11 reextracted samples were also pre-heated with 120°C for 8h and subsequently [14]C-dated (see Table 1, samples with ETH Nr's.). For direct comparison of the effectiveness of the pre-heating approach, *n*-alkane fractions of 4 reextracted samples were taken from the same depths of the investigated FSPS as those from the pre-test that were formerly not heated.

The *n*-alkanes were identified and quantified using a gas-chromatograph (Agilent 7890 with an Agilent HP5MS column) equipped with a flame ionization detector (GC-FID). For identification and quantification, external *n*-alkane standards ($C_{21}$ – $C_{40}$) were run with each sequence.

### 2.3.2 [14]C measurements

The [14]C analyzes were carried out at both the LARA AMS Laboratory of the University of Bern (Szidat, 2014) and the LIP of the ETH Zurich, Switzerland.

For compound-class dating, purified non-heated and pre-heated *n*-alkane fractions were transferred with Dichlormethane into tin capsules (3.5 * 5.5 * 0.1 mm). [14]C-dating was performed on the Mini Carbon Dating System (MICADAS) AMS coupled online to an Elementar Analyzer (Wacker et al., 2010; Ruff et al., 2011). Results are reported as fraction modern ($F^{14}C$), which is the acitivity ratio of a sample related to the modern reference material Oxalic Acid II after subtracting the background signal. The *n*-alkanes of the pre-test (step 1) were analyzed at the LARA AMS (BE Nr's.). $F^{14}C$ results from the LARA AMS were corrected for cross (carry over from sample to sample) and constant contamination (carbon mass and $F^{14}C$ of the tin caps) according to the contamination drift model of Salazar et al. (2015): For constant contamination, 10 combined tin capsules were measured which yielded 0.43 μg C for a single cap with $F^{14}C$ values of 0.759.





The reextracted and pre-heated *n*-alkane fractions (step 2) were analyzed at the LIP AMS (ETH Nr's.). $F^{14}C$ results were corrected for constant contamination according to Welte et al. (2018) with a fossil *n*-alkane standard ($C_{28}$ with $F^{14}C = 0$) and a modern *n*-alkane standard ($C_{32}$ with $F^{14}C = 1.073$). Constant contamination correction yielded 4.3 µg C for a single cap, with $F^{14}C$ values of 0.895.

All $^{14}C$-ages were calibrated to cal. years BP (95.4% range) with the IntCal13 calibration curve (Reimer et al., 2013) using OxCal (Ramsey, 2009). $^{14}C$ results were reported following the conventions of Millard (2014).

## 3 Results and Discussion

All obtained $^{14}C$-ages are found in Table 1.

### 3.1 Heating experiments

Fig. 4A shows the homologue distribution of petrogenic *n*-alkanes derived from a Jurassic black clay shale sample from the upper Alazani sub-catchment (Fig. 2C). These are present with similar amounts at all chain-lengths ($C_{21}$-$C_{35}$) and do not show a distinct odd-over-even predominance (OEP). These $^{14}C$-dead *n*-alkanes also contribute to the sedimentary leaf wax *n*-alkanes in our FSPS, and are exemplarily shown as the assumed maximal petrogenic contribution for the *n*-alkane homologue distribution of non-heated sample Ala 425 I in Fig. 4B. During the pre-test, this sample gave a $^{14}C$-age of 12.4-13.3 cal. ka BP

(95.4%), what is much older than the timing of sedimentation obtained from the charcoal $^{14}C$-ages (~5 to 4 cal. ka BP). To at least partly reduce the contamination with petrogenic *n*-alkanes, we reextracted this sample and applied the heat treatment of 120°C for 8 h before $^{14}C$-measurement. The *n*-alkane homologue distribution of the reextracted and heated sample Ala 415 II (Fig. 4C) demonstrates that the short-chain *n*-alkanes and thus a significant amount of the petrogenic *n*-alkanes were effectively removed by this procedure. Consequently, the resulting $^{14}C$-age of 6.4-7.9 cal. ka BP (95.4%) was much younger than the age

obtained without pre-heating (Fig. 4C and 5).

The $^{14}C$-ages of the other samples that were also measured both before and after the removal of the short-chain *n*-alkanes by heating were: For sample Ala 225 5.8-6.3 cal. ka BP (95.4%) (Ala 225 I) and 3.1-4.2 cal. ka BP (95.4%) (Ala 225 II), for sample Ala 290 7.6-8.2 cal. ka BP (95.4%) (Ala 290 I) and 5.8-7.2 cal. ka BP (95.4%) (Ala 290 II), and for sample Ala 505 15.7-16.8 cal. ka BP (95.4%) (Ala 505 I) and 6.5-8.6 cal. ka BP (95.4%) (Ala 505 II), respectively (Table 1; Fig. 5).

The results of our heating experiments show that a simple heat treatment effectively removes the short-chain *n*-alkanes that contain a significant petrogenic component. However, we have to mention that no extensive heating experiments were carried out for the removal of short-chain *n*-alkanes prior to this study, and that we only tested at 3 different temperatures (100, 110 and 120°C) for 8h. We found that most of the short-chain *n*-alkanes could best be removed with 120°C for 8h, although a slight proportion of the longer-chains were also removed. We also have to mention that possible fractionation during heating could

potentially lead to an enrichment of the heavier $^{14}C$-isotope, but such a fractionation effect should be negligible because the $F^{14}C$ results were generally corrected for mass-dependent fractionation by the $^{13}C$-isotopes. Therefore, the applied heat





treatment is an effective pre-treatment to derive a more homogenuous leaf wax *n*-alkane signal for compound-class [14]C-dating in general, and especially in environmental settings where petrogenic OC and *n*-alkanes occur. Because of the partly removed petrogenic contribution, the [14]C-age of a heated sample is generally closer to its "true" leaf wax *n*-alkane [14]C-age. However, since petrogenic *n*-alkanes are also present at the longer chains $\geq C_{25}$ that mostly originate from leaf waxes and serve as

paleoenvironmental proxies (see Fig. 4), this pre-treatment cannot completely remove the petrogenic contribution.

### 3.2 *n*-Alkane [14]C-ages from the FSPS along the upper Alazani River

In the following, we will only focus on the 11 *n*-alkane samples from the investigated FSPS that were pre-heated with 120°C for 8 h. *n*-Alkanes were present in all these samples after heating with values ranging between 27 to 60 µg carbon per *n*-alkane fraction, and they contained enough carbon for robust [14]C-measurements. Their $F^{14}C$-values range from $0.7865 \pm 0.0110$ to

$0.3544 \pm 0.0174$, what corresponds to calibrated calendar ages from 1.6-2.2 cal. ka BP (95.4%) to 8.4-10.7 cal. ka BP (95.4%) (Table 1).

Calibrated calendar ages for the *n*-alkanes from (paleo)soils and fluvial sediment layers are shown in Fig. 6. Compared to the independent charcoal-based chronology of the sequence (Suchodoletz et al., 2018), the *n*-alkane [14]C-ages show variable age offsets over the FSPS, i.e. they are generally older. The *n*-alkane [14]C-ages of intensively developed paleosols and the recent

soil show generally lower age offsets (between 0 and 1.3 ka) than those from less intensively developed paleosols (up to 3.3 ka) (Fig. 6). The age offsets of the *n*-alkanes from fluvial sediment layers are generally older than those from (paleo)soils, and increase in the upper part of the FSPS (Fig. 6): Age offsets in the lower part range between ~2.3 and 3.5 ka, and increase up to ~6 ka in the upper part.

### 3.3 Estimation and correction for petrogenic *n*-alkanes

Our pre-heating of the *n*-alkane fractions with 120°C for 8h effectively removed the short-chained petrogenic *n*-alkanes ($<C_{25}$) derived from Jurassic black clay shales in the upper Alazani sub-catchment. However, such petrogenic *n*-alkanes are also present at the longer chains ($C_{25}$-$C_{35}$), where they underlie the leaf wax-derived *n*-alkanes and do not show a distinct OEP (see Fig. 4C). Our pre-heating can therefore not fully remove all petrogenic *n*-alkanes. To correct for this underlying petrogenic contribution, we propose a simple correction procedure that uses a constant correction factor: A measured Jurassic black clay

shale sample from the upper Alazani sub-catchment yielded an averaged concentration of 0.007 µg*g$^{-1}$ per single *n*-alkane compound (see Fig. 4A). Assuming that this is the maximal possible concentration of petrogenic *n*-alkanes in sediment samples of our FSPS, and that this equally underlies both the even and odd leaf wax compounds from fluvial sediments and (paleo)soils, the proportion of petrogenic *n*-alkanes for each sample can be quantified and calculated as follows:

$$\text{petrogenic contribution (\%)} = \frac{0.007\ \mu g*g^{-1} * \text{sum of chains}}{n\text{-alkane concentration } \mu g*g^{-1}} *100 \tag{1}$$





The sum of chains is the number of $n$-alkane chains from $C_{25}$ to $C_{35}$, and the $n$-alkane concentration is the sum of concentration from $C_{25}$ to $C_{35}$. In cases where not all shorter chains ($<C_{25}$) could completely be removed from the sample by heating, they will be included in the calculation.

The calculated petrogenic contribution (%) that is assumed to be $^{14}$C-dead (i.e. has a $F^{14}C$ value = 0) can then be subtracted from the measured $F^{14}C$-value to derive the petrogenic corrected $F^{14}C$-value ($F^{14}C_{petro\ corr}$):

$$F^{14}C_{petro\ corr} = F^{14}C_{measured} + (F^{14}C_{measured} * petrogenic\ contribution * 0.01) \tag{2}$$

The obtained petrogenic corrected $F^{14}C$-values were then calibrated with IntCal13 to yield calendar ages again. These ages indicate the maximal possible petrogenic contribution in the sediment samples from our FSPS. Because of dilution of petrogenic $n$-alkanes in the sediment samples by not black clay shale-derived sediment material and/or the slight loss of long-chain $n$-alkanes during heating (see discussion 3.1), the "true" leaf wax $n$-alkane ages lie in-between the heated and petrogenic-corrected ages.

As illustrated in Fig. 7, the maximal possible petrogenic contribution from petrogenic $n$-alkanes can be quantified and corrected for by this approach. Maximal possible petrogenic contributions are relatively high with ~0.8 to 2.0 ka throughout the investigated FSPS. Compared to the fluvial sediments, the (paleo)soils show lower contributions of petrogenic $n$-alkanes. Due to generally higher total $n$-alkane concentrations in the upper part of the FSPS, the relative proportion of the maximal possible petrogenic contribution generally decreases from bottom to top (Fig. 7).

### 3.4 *In-situ* leaf wax $n$-alkane formation versus pre-aging and reworking

Since the petrogenic $n$-alkanes could effectively be removed by heating and correction for maximal petrogenic contributions, the remaining $n$-alkanes should mostly derive from leaf waxes. Accordingly, their age offset to the sediment layers in the FSPS wherein they are buried became smaller. However, the age offsets still reach up to several millennia and vary over the FSPS, and have to be regarded as minimal age offsets since the correction for petrogenic contributions gave maximal possible values. These offsets generally differ between (paleo)soils and fluvial sediment layers (Fig. 7).

*Fluvial sediment layers*

Calculated petrogenic corrected minimal age offsets for the leaf wax $n$-alkanes from fluvial sediment layers range between ~0.4 to ~5.0 ka. They are distinctively larger in the upper part of the FSPS, i.e. in the fluvial sediment layers above Ahb1, Ahb2 and Ahb3. In the lower part of the FSPS below Ahb3 and Ahb4, corrected age offsets strongly decrease and are only slightly off with the timing of sedimentation (Fig. 7). We interpret these variable age offsets to be caused by different degrees of pre-aging and reworking of OC and leaf wax $n$-alkanes, which can mainly be caused by three different effects: (i) Different degrees of pre-aging and reworking of OC and leaf wax $n$-alkanes: The proportion of recent to sub-recent versus pre-aged and reworked OC and leaf wax $n$-alkanes in FSPS is primarily controlled by the intensity of physical erosion in the catchment (Hilton et al., 2012; Smith et al., 2013; Galy et al., 2015). Thus, larger $^{14}$C-age offsets in the upper part of our FSPS indicate



more intensive and profound erosion in the upper Alazani sub-catchment during the last 4 cal. ka BP, leading to a relatively large-scale mobilization of pre-aged OC and leaf wax *n*-alkanes compared with recent to sub-recent ones. In contrast, smaller age offsets in the lower part of the FSPS indicate less intensive and profound erosion before ca. 4 cal. ka BP, while a larger relative amount of recent to sub-recent OC and leaf wax *n*-alkanes was mobilized (see Fig. 7). (ii) Different durations of *in-situ* soil formation and the according buildup of OC and leaf wax *n*-alkanes in catchment soils: In case that they were not occasionally eroded recent soils continuously built up during the Holocene, and thereby the mean age of soil OC and leaf wax *n*-alkanes became successively older (Smittenberg et al., 2006; Gierga et al., 2016). Therefore, OC and *n*-alkanes that were eroded after a longer time of soil development during the late Holocene should exhibit larger mean age offsets than those that were eroded during the early or middle Holocene, i.e. after a shorter time of soil development. Thus, even in case of a constant proportion of recent to sub-recent versus pre-aged and reworked OC and leaf wax *n*-alkanes in the sediments, a systematic increase of their age throughout the Holocene should be expected. (iii) Sediment (dis)connectiviy (Leithold et al., 2006): Disconnectivity in a hydrological catchment describes blockages like sediment sinks and storages that temporarily interrupt longitudinal, lateral and vertical sediment delivery (Fryirs, 2013), leading to increasing OC and leaf wax *n*-alkane age offsets. However, because of its relatively small size of ~1.100 km$^2$ and the steep averaged slopes of ~20° (Fig. 2B), the upper Alazani sub-catchment has a high sediment connectivity. This is also demonstrated by the general absence of fine-grained overbank sediments upstream of the confluence of uppermost Alazani and Ilto River. Therefore, most of the fine-grained material in the investigated FSPS must have been eroded relatively shortly before final deposition, i.e. without a significant time lag between both processes (Bliedtner et al., 2018a; Suchodoletz et al., 2018).

Nevertheless, since disconnectivity effects in the upper Alazani sub-catchment are negligible and although the continuous buildup of OC and leaf wax *n*-alkanes in catchment soils can contribute to the age offsets, strongly increasing age offsets after ~4 cal. ka BP are most likely a result of increased erosion due to intensified regional anthropogenic activity since ~4.5 ka BP (Akhundov, 2004).

*(Paleo)soils*

For the intensively developed (paleo)soils, petrogenic-corrected leaf wax *n*-alkane ages show no minimal age offsets and thus fall into the period of soil formation (Fig. 7). In contrast, corrected minimal age offsets of the less intensively developed paleosols Ahb 4 and Ahb 2 strongly increase up to ~2.4 ka (Fig. 7). Thus, in these weakly developed paleosols the *in-situ* leaf wax *n*-alkane signal from local biomass decomposition is obviously still much stronger biased by inherited pre-aged and reworked leaf wax *n*-alkanes from their fluvial parent material compared to the intensively developed (paleo)soils. In the latter, the longer duration of soil development must have constantly incorporated locally-derived leaf wax *n*-alkanes that fully overprinted the formerly deposited leaf wax *n*-alkanes from their fluvial parent material.





Taken together, whereas relatively large age offsets between leaf wax *n*-alkane formation and deposition can occur in less intensively developed paleosols and fluvial sediment layers, leaf wax *n*-alkane ages from intensively developed (paleo)soils reflect most reliably the timing of their formation.

**3.5 Implications for leaf wax *n*-alkane-based paleoenvironmental reconstructions from our FSPS**

*Fluvial sediment layers*

The interpretation of catchment-derived leaf wax *n*-alkanes from the fluvial sediment layers is very challenging, as they show variable minimal age offsets of ~0.3 to 4.7 ka over the investigated FSPS. It appears that all dated leaf wax *n*-alkanes from fluvial sediment layers in the FSPS must have formed during a similar period in the middle Holocene between ~8 and ~5.6 cal. ka BP (Fig. 8). However, the start of their formation could also have been some centuries earlier, since the petrogenic

correction only gave minimal values (Fig. 7). A dominance of deciduous tree/shrub-derived leaf wax *n*-alkanes in fluvial sediments between ca. 390 and 170 cm indicates that larger parts of the upper Alazani sub-catchment must have been forested during the middle Holocene (Fig. 7), and grass/herb-derived leaf wax *n*-alkanes from the lower (between ca. 575 and 480 cm) and uppermost part of the FSPS (above ca. 130 cm) with similar ages indicate that other parts of the catchment must have been covered by grass/herb vegetation during the same period (Fig. 8). Therefore, catchment-derived leaf wax *n*-alkanes were

obviously not constantly eroded relatively shortly after their formation, as was formerly suggested by Bliedtner et al. (2018a). Instead, large-scale erosion of leaf wax *n*-alkanes with similar middle Holocene ages must have occurred in different parts of the sub-catchment during different periods (Fig. 8).

*(Paleo)soils*

Our results show that the petrogenic-corrected leaf wax *n*-alkane [14]C-ages from the intensively developed (paleo)soils Ah,

Ahb1, Ahb5 and Ahb6, that were all formed for at least 1 ka, generally agree with the independent [14]C-ages from charcoal (Fig. 7). That means that the respective leaf wax proxies from such soils, e.g. the commonly used ACL, OEP and stable hydrogen/carbon isotopes, give reliable paleoenvironmental information about the *in-situ* formed leaf wax *n*-alkane signal. Thus, since *in-situ* formed leaf waxes obviously dominate the *n*-alkane signal in the intensively developed (paleo)soils of the FSPS in the upper Alazani valley that indicate high grass/herb percentages at the studied site throughout the Holocene

(Bliedtner et al., 2018a, Figs. 7 and 8), this is chronologically not biased by pre-aging and reworking effects. While the natural potential vegetation in the upper Alazani lowlands are elm-oak-vine forests rather than grassland (Connor and Kvavadze, 2008), deciduous trees have been reported by pollen analyses from buried soil profiles for the neighboring Iori lowlands before 5 cal. ka BP (Gogichaishvili, 1984; location see Fig. 1 and record Fig. 8). Therefore, increased anthropogenic activity is the most likely cause for the observed grass/herb dominance in the (paleo)soils at the studied site for most of the Holocene

(Bliedtner et al., 2018a). Anthropogenic influence at the studied site is documented by archeological artefacts in paleosol Ahb6 since ~8 cal. ka BP (Fig. 3), and regional anthropogenic activity intensified since ~4.5 cal. ka BP (Akhundov, 2004; see Fig.





8). Compared with the intensively developed (paleo)soils, relatively large offsets between the petrogenic-corrected leaf wax *n*-alkane [14]C-ages and the independent charcoal [14]C-ages are found for the less intensively developed paleosols Ahb4 and Ahb2. This is caused by the larger proportion of pre-aged and reworked leaf wax *n*-alkanes from their fluvial parent material compared with the *in-situ* formed signal. This is further demonstrated by the fact that the minimal age offset of ~0.7 ka for

5    paleosol Ahb4 that had formed for ca. 400 years is significantly smaller than that of ~2.2 ka for paleosol Ahb2 that had only formed for ca. 50 years (Fig. 7). However, the results from the weakly developed paleosols do not affect the picture of a general dominance of grass/herb-vegetation at the investigated site throughout the Holocene (Fig. 8).

*Paleoenvironmental reconstruction*

Although our leaf wax *n*-alkane [14]C-ages do not support the former leaf wax *n*-alkane-based paleovegetation interpretation of

10    the study of Bliedtner et al. (2018a) in every detail, they confirm its main findings:

- Despite other results from a neighboring pollen archive near Sagarejo (Gogichaishvili, 1984; location see Fig. 1 and record Fig. 8), the upper Alazani floodplain has been dominated by grass/herb vegetation throughout the Holocene. This was most probably caused by anthropogenic activity.

- At least parts of the upper Alazani sub-catchment were covered with deciduous trees/shrubs during the middle

15        Holocene between ~8 and ~5.6 cal. ka BP. No older ages were determined from leaf wax *n*-alkanes derived from this type of vegetation. Therefore, it is very likely that this marks the beginning of post-glacial reforestation in the upper Alazani sub-catchment, although the start of reforestation could also have occurred some centuries earlier since the petrogenic correction only gave minimum values. This corroborates former pollen data that suggest a delayed regional post-glacial reforestation compared with western and central Europe between ca. 9 to 6 cal. ka BP (Lake Urmia:

20        Bottema, 1986; Lake Van: Litt et al., 2009; Lake Paravani: Messager et al., 2013; locations see Fig. 1, and record from Lake Paravani Fig 8).

- Middle Holocene leaf wax *n*-alkanes that were deposited before ca. 4 cal. ka BP dominantly originate from grass/herb vegetation. In contrast, most middle Holocene leaf wax *n*-alkanes with similar ages that were deposited after that time originate from deciduous tree/shrubs. This suggests the start of large-scale erosion around ca. 4 cal. ka BP in those

25        parts of the upper Alazani sub-catchment that were covered by deciduous forest vegetation. Furthermore, given larger age offsets between biomarker formation and deposition than before, this suggests more profound erosion processes with a relatively larger mobilization of pre-aged leaf wax *n*-alkanes since ca. 4 cal. ka BP. This finding agrees with the observation that settlement in higher altitudes of the Greater Caucasus started since ca. 4.5 cal. ka BP (Akhundov, 2004; Fig. 8).


## 4 Conclusions

During this study, we directly dated leaf wax *n*-alkanes from a fluvial sediment-paleosol sequence (FSPS) along the upper Alazani River in eastern Georgia by compound-class $^{14}$C-dating to investigate their potential for paleoenvironmental reconstructions. Our study gave the following results:

- Pre-heating of the *n*-alkane fraction with 120°C for 8h before compound-class $^{14}$C-dating effectively removed the short-chain *n*-alkanes ($<C_{25}$). These included a part of the petrogenic *n*-alkane contribution from Jurassic black clay shales from the upper catchment.

- Remaining petrogenic contributions of the long-chain *n*-alkanes ($\geq C_{25}$) were estimated and corrected for by applying a simple constant petrogenic correction factor that is based on *n*-alkane concentrations in a Jurassic black clay shale

sample from the upper Alazani sub-catchment. The corrected *n*-alkane $^{14}$C-ages from the FSPS were younger than without correction, and gave the "true" leaf wax *n*-alkane age information.

- A part of the petrogenic-corrected leaf wax *n*-alkanes still showed relatively large age offsets between their formation and final deposition into the FSPS, indicating different degrees of pre-aging and reworking: While there is no offset for leaf wax *n*-alkanes from intensively developed (paleo)soils what indicates a dominance of *in-situ* produced local

leaf wax *n*-alkanes, the offsets for leaf wax *n*-alkanes from less intensively developed paleosols are much larger. This can possibly be explained with a larger relative proportion of inherited leaf wax *n*-alkanes from their fluvial parent material. Accordingly, the offsets in fluvial sediment layers are even larger and show values up to several thousand years. Since all dated leaf wax *n*-alkanes from fluvial sediment layers show similar middle Holocene ages between ~8 and ~5.6 cal. ka BP, the age offsets generally increase towards the top of the sequence, indicating a greater relative

proportion of pre-aged and reworked compared with recent to sub-recent leaf wax *n*-alkanes in fluvial sediments deposited after ca. 4 cal. ka BP.

- Leaf waxes from intensively developed (paleo)soils showed a dominance of grass/herb vegetation at the studied site throughout the Holocene. This was most likely caused by anthropogenic influence since ~8 cal. ka BP. Middle Holocene *n*-alkanes from fluvial sediment layers in different parts of the FSPS indicate both deciduous trees/shrubs

and grasses/herbs in the upper Alazani sub-catchment since at latest ~8 cal. ka BP. Similar with former regional studies, this indicates a delayed post-glacial reforestation in the upper Alazani sub-catchment compared with western and central Europe since ca. 9 - 8 cal. ka BP. Unlike those that were deposited before, the leaf wax *n*-alkanes that were deposited since ca. 4 cal. ka BP showed a dominant origin from deciduous trees/shrubs, which obviously indicates the start of large-scale erosion in deciduous forest-covered parts of the sub-catchment since that period.

Our results demonstrate that compound-class $^{14}$C-dating of *n*-alkanes in FSPS is important to investigate their age and origin, since varying proportions of both local and catchment-derived leaf wax *n*-alkanes are found in these archives. Therefore, this step is a mandatory precondition for robust leaf wax *n*-alkane-based paleoenvironmental reconstructions from FSPS, but also from other kinds of sediment archives with hydrological catchments such as lake and marine sediments. Generally, for leaf



wax *n*-alkane-based paleoenvironmental studies in such sediment archives we recommend to select (i) catchments without carbon-rich sediment rocks containing petrogenic *n*-alkanes, and (ii) relatively small catchments with short mean transfer times between leaf wax *n*-alkane formation and deposition.

## Acknowledgments

This project was financially supported by the Swiss National Science Foundation (project P00P2-150590). We thank Ulrich Göres (Dresden) and Giorgi Merebashvili (Tbilisi) for their help during fieldwork.

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



**Table 1:** *n*-Alkane [14]C-ages from the investigated FSPS along the upper Alazani River including treatment, carbon mass, fraction modern values (F[14]C), uncalibrated [14]C-ages and calibrated [14]C-ages as calibrated age ranges in cal. BP (95.4%) with the calibrated median age. The *n*-alkane samples from the pre-test are marked with I, whereas the corresponding reanalyzed *n*-alkane samples with II for direct comparison. Colored rows indicate *n*-alkane samples from (paleo)soils.

| Lab. code | Sample label | Pre-treatment | Depth (cm) | Mass (ug) | F[14]C | Uncalibrated ages | Calibrated age ranges (cal. BP) | Calibrated median age (cal. BP) |
|---|---|---|---|---|---|---|---|---|
| BE-4788.1.1 | Ala 25 I | Heated | 25 | 30 | 0.7865 ± 0.0110 | 1929 ± 112 | 1573-2150 (95.4%) | 1877 |
| ETH-81313.1.1 | Ala 100 | Heated | 100 | 49 | 0.4490 ± 0.0175 | 6432 ± 311 | 6660-7934 (95.4%) | 7304 |
| BE-4792.1.1 | Ala 225 I | Not heated | 225 | 55 | 0.5199 ± 0.0066 | 5255 ± 101 | 5756-6282 (95.4%) | 6046 |
| ETH-81312.1.1 | Ala 225 II | Heated | 225 | 60 | 0.6583 ± 0.0159 | 3359 ± 193 | 3083-4150 (95.4%) | 3623 |
| ETH-81311.1.1 | Ala 280 | Heated | 280 | 54 | 0.3544 ± 0.0174 | 8334 ± 390 | 8390-10264 (95.4%) | 9310 |
| BE-4793.1.1 | Ala 290 I | Not heated | 290 | 47 | 0.4148 ± 0.0082 | 7069 ± 158 | 7595-8191 (95.4%) | 7895 |
| ETH-81310.1.1 | Ala 290 II | Heated | 290 | 55 | 0.4974 ± 0.0164 | 5610 ± 263 | 5769-7156 (95.3%) | 6427 |
| ETH-81309.1.1 | Ala 315 | Heated | 315 | 41 | 0.3598 ± 0.0205 | 8212 ± 452 | 8170-10371 (95.4%) | 9181 |
| ETH-81308.1.1 | Ala 390 | Heated | 390 | 47 | 0.4050 ± 0.0186 | 7261 ± 366 | 7468-8987 (95.4%) | 8120 |
| BE-4795.1.1 | Ala 425 I | Not heated | 425 | 47 | 0.2564 ± 0.0065 | 10935 ± 203 | 12430-13255 (95.4%) | 12850 |
| ETH-81307.1.1 | Ala 425 II | Heated | 425 | 37 | 0.4553 ± 0.0208 | 6319 ± 363 | 6415-7927 (95.4%) | 7183 |
| BE-4796.1.1 | Ala 505 I | Not heated | 505 | 37 | 0.1866 ± 0.0048 | 13488 ± 206 | 15685-16930 (95.4%) | 16257 |
| ETH-81306.1.1 | Ala 505 II | Heated | 505 | 27 | 0.4351 ± 0.0271 | 6685 ± 494 | 6494-8584 (95.4%) | 7565 |
| ETH-81305.1.1 | Ala 545 | Heated | 545 | 31 | 0.4253 ± 0.0243 | 6867 ± 454 | 6749-8715 (95.4%) | 7753 |
| ETH-81303.1.1 | Ala 625 | Heated | 625 | 36 | 0.3782 ± 0.0220 | 7811 ± 463 | 7743-9887 (95.4%) | 8748 |



**Figures**

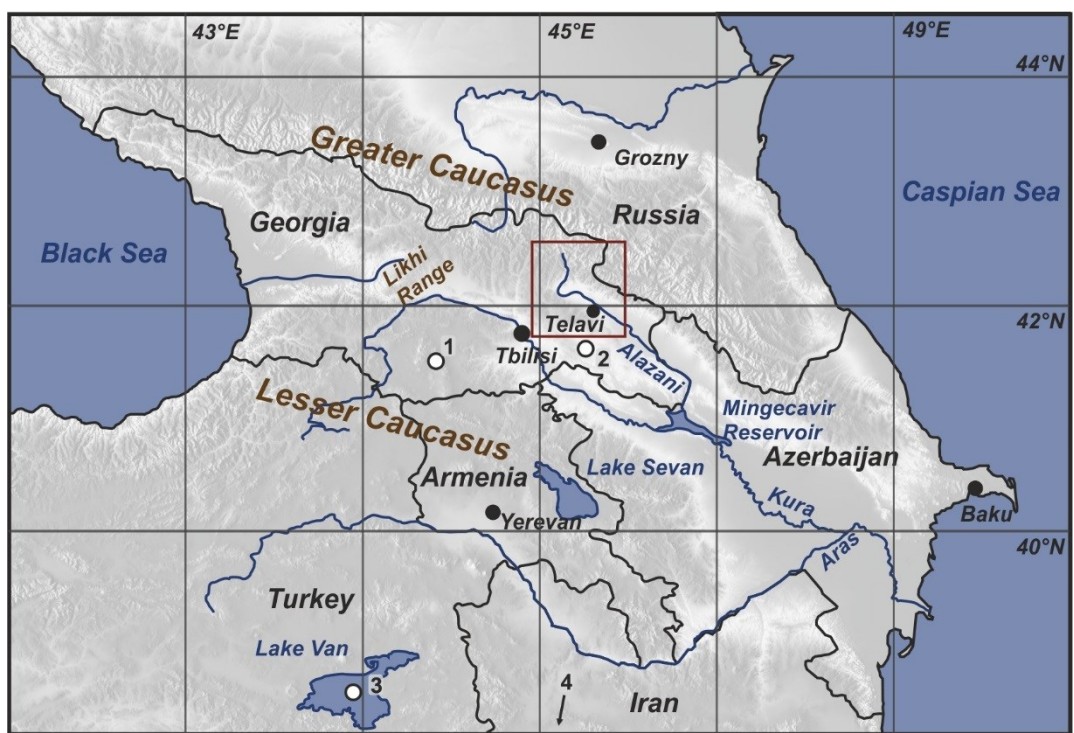

Figure 1: Overview of the Caucasus region. The red rectangle marks the study area in the upper Alazani valley. Regional pollen records that are used for comparison in this study are marked with a white dot: 1 = Lake Paravani (Messager et al., 2013), 2 = Sagarejo sediment section (Gogichaishvili, 1984), 3 = Lake Van (Litt et al., 2009), 4 = Lake Urmia (Bottema, 1986).

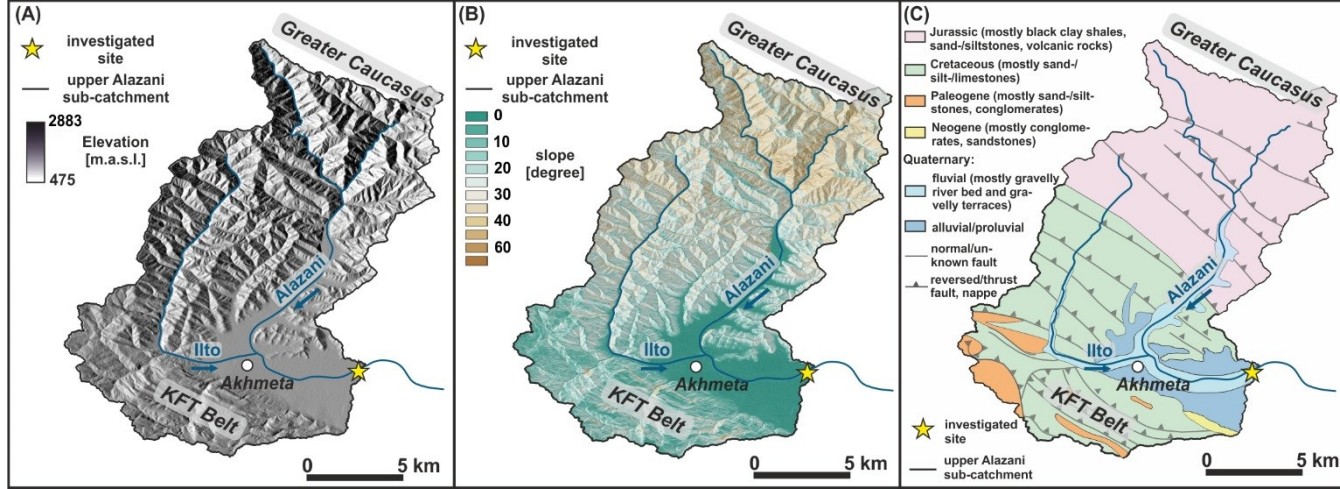

Figure 2: (A) Digital elevation model of the upper Alazani sub-catchment (SRTM30-DEM). (B) Slope map of the upper Alazani sub-catchment. (C) Geological map of the upper Alazani sub-catchment (simplified after Gamkrelidze, 2003). KFT Belt = Kura-fold-thrust-belt.




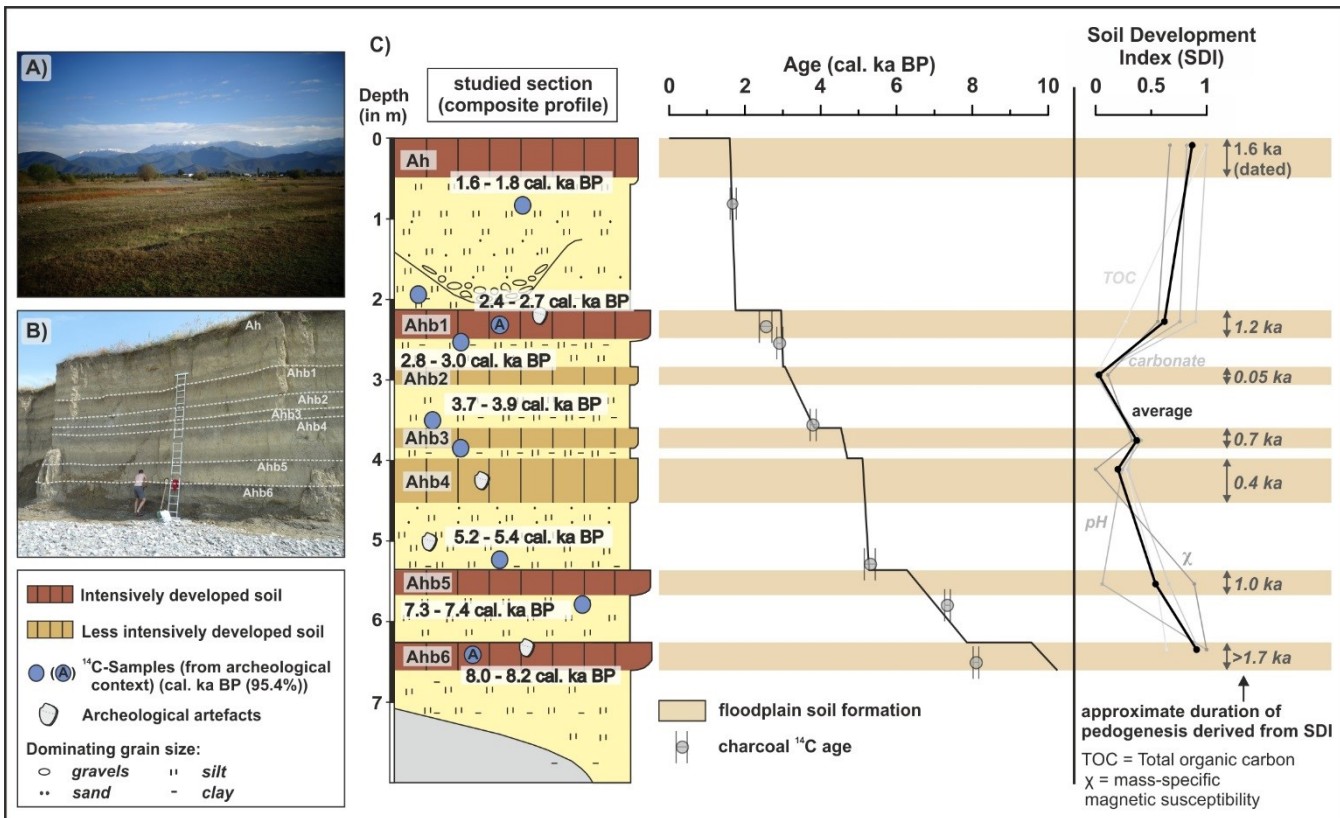

Figure 3: A) Photo of the active modern floodplain next to the investigated FSPS in the front, and the mountainous upper Alazani catchment in the southern Greater Caucasus in the back. B) Photo of the investigated FSPS with highlighted (paleo)soils. C) Schematic stratigraphy of the investigated FSPS (left) with the age-depth model based on $^{14}$C-datings of charcoal pieces (center), and approximate durations of soil formation based on a soil development index (SDI) (right; Suchodoletz et al., 2018). The two charcoal samples that overestimated their burial ages were excluded from the age-depth model. Charocal $^{14}$C-ages are given as calibrated age ranges in cal. ka BP (95.4%) with the calibrated median age.


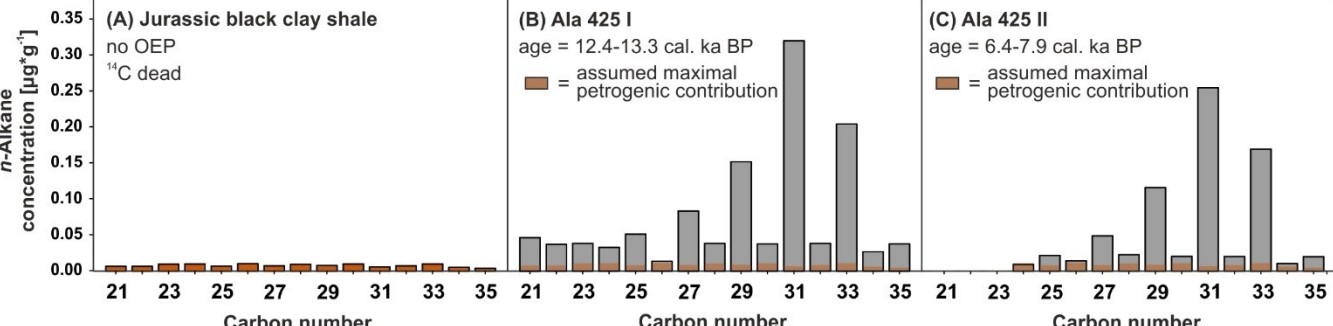

**Figure 4:** *n*-Alkane chain-length distributions in µg*g⁻¹ sediment for (A) a Jurassic black clay shale sample from the upper Alazani sub-catchment, (B) FSPS sediment sample Ala 425 I without heating, and (C) reextracted FSPS sediment sample Ala 425 II after heating with 120°C for 8 h. *n*-Alkane $^{14}$C-ages for samples Ala 425 I and II are given as age ranges in cal. ka BP (95.4%).

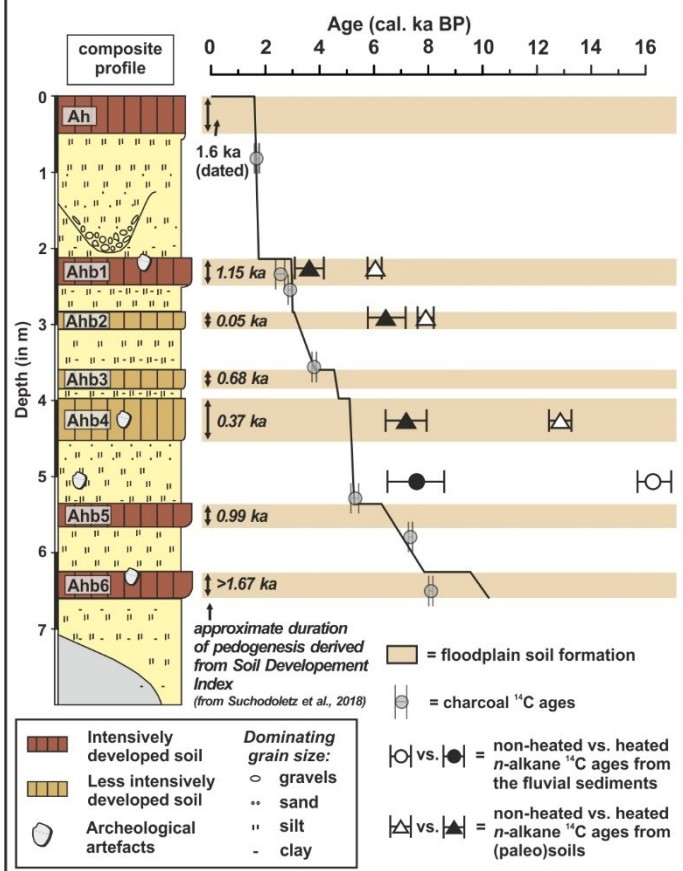

**Figure 5:** Comparison of non-heated *n*-alkane $^{14}$C-ages from the pre-test and heated *n*-alkane $^{14}$C-ages that were reextracted from the same samples of the investigated FSPS and heated with 120°C for 8h. For comparison, the independent age model based on $^{14}$C-dated charcoal pieces and the Soil Development Index (SDI) is also shown (Suchodoletz et al., 2018). $^{14}$C-ages are given as calibrated age ranges in cal. ka BP (95.4%) with the calibrated median age.



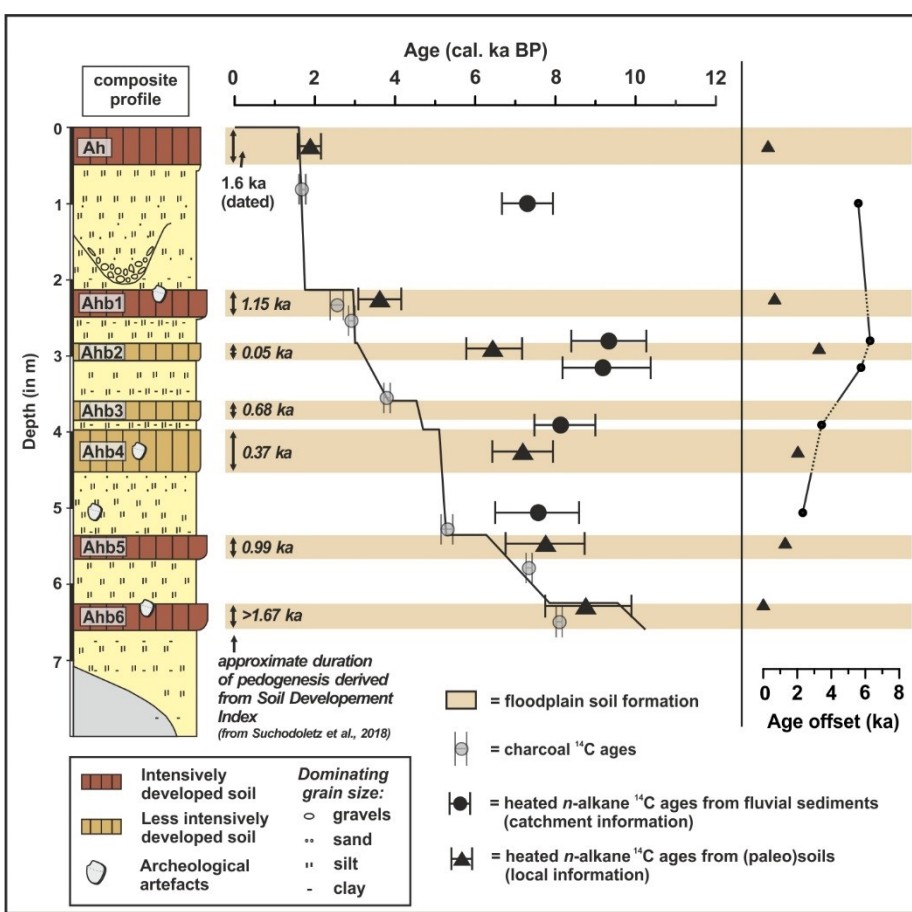

**Figure 6: Chronostratigraphy of the investigated FSPS with *n*-alkane [14]C-ages after heating with 120°C for 8 h from (paleo)soils and fluvial sediment layers. [14]C-ages are given as calibrated age ranges in cal. ka BP (95.4%) with the calibrated median age.**



**Figure 7: Chronostratigraphy of the investigated FSPS with leaf wax *n*-alkane [14]C-ages corrected for maximal possible petrogenic contributions from catchment-derived Jurassic black clay shales. Contributions of grasses/herbs have been formerly reported by Bliedtner et al. (2018a). Datapoints showing the percentages of grasses/herbs derived from Bliedtner et al. (2018a) that do not have a direct [14]C-age information from this study are plotted in a transparent way, and datapoints with such kind of [14]C-age information are shown without transparency.**



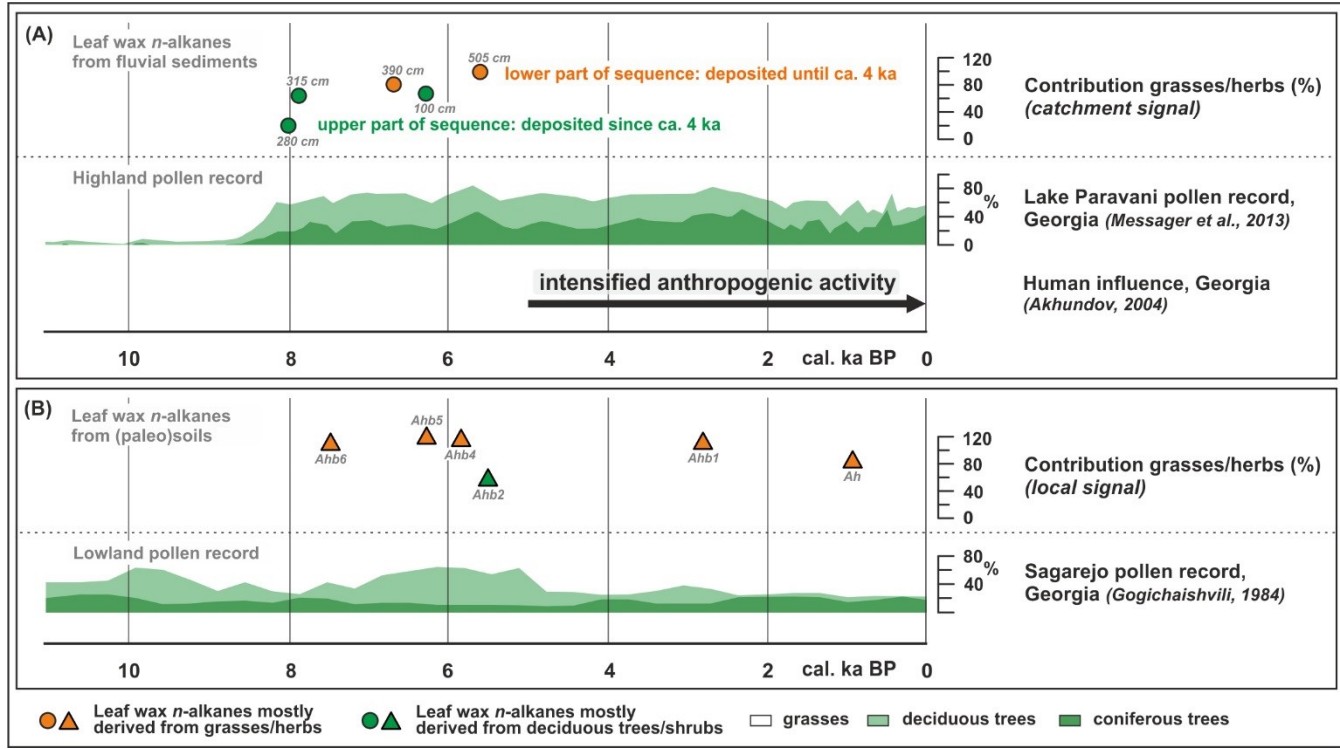

**Figure 8: A) Leaf wax *n*-alkane record from fluvial sediments from the investigated FSPS compared with the regional highland pollen record from Lake Paravani (Messager et al., 2013; location see Fig. 1), and with increasing human activity in the region (Akhundov, 2004). B) Leaf wax *n*-alkane record from (paleo)soils from the investigated FSPS compared with the lowland Sagarejo sediment section (Gogichaishvili, 1984; locations see Fig. 1). Contributions of grasses/herbs derived from Bliedtner et al. (2018a; see highlighted samples in Fig. 7) are plotted with the corresponding [14]C-ages from this study.**