# Peer review of "Age and origin of leaf wax *n*-alkanes in fluvial sediment-paleosol sequences, and implications for paleoenvironmental reconstructions"

_Hydrology and Earth System Sciences, 2019_

## Referee Comment (RC1) · Ulrich Hanke (Referee) · 23 Aug 2019

Marcel Bliedtner and collaborators investigate a paleosol sequence of the upper Alazani river valley in Eastern Georgia with molecular-level radiocarbon records of long-chain leaf waxes. The aim of this study is to differentiate between sources of n-alkanes in sedimentary deposits via radiocarbon isotope mass balance calculations of two sources: petrogenic (shale) and pre-aged (catchment).

The authors employ compound-class 14C measurements of long-chain leaf waxes (C

$\geq$ 25) after removing the shorter chain lengths (C < 25) during the laboratory analyses to reduce the impact of petrogenic C (and reduce uncertainties from microbial reworking). However, even long chain leaf waxes contain a fraction of petrogenic carbon for which the authors introduce a factor and correct their 14C values to then discuss the pre-aging. Further, they translate their extracted 14C values to obtain calendar years yet hardly discuss the additional sensitivities introduced during "calibrating" with the 14C reference Intcal13. Or do the authors speak about 14C years? The former can be tricky because the paleo-fluvial sedimentary sequence seems to have features of recent biological activity and contemporary carbon that can complicate any attempt of absolute age dating.

The excellent combination of molecular-level geochemical tools to trace the fate of carbon in past fluvial deposits is of great interest to earth scientists from various disciplines and, to my understanding, well suited for publication in HESS. The manuscript is scientific sound, entails adequate illustrations and details. However, the current version could benefit from (i) improve clarity in several sections (sentence length and perhaps language); (ii) some clear statement on catchment-wide molecular-level 14C data that consists of an age distribution of 14C rather than a single value; (iii) a statement on the informative value on ACL vs isotopes (13C and 14C); (iv) discussion of data in the light of contemporary and pre-aged carbon; (v) some more informative details on potentials and limits on the geochemical constraints of molecular environmental 14C data compared to conventional 14C dating (in archeology). Probably some of my comments may already be included in the manuscript and might become clear after some careful checking/shortening of sentences and the addition of some clear statements. Overall a great study.

Some specific comments: 1,17: is 'direct' the correct term since you clean your fractions prior to 14C analyses?

1,19-20: 'in-situ signal from local biomass': do you mean contemporary or on-site from litter fall?

[Figure]

1, 30: 'were estimated': how do you determine the petrogenic C contributions? If only estimated, you may need to add some more details on the rationale and the precision of your approach

2, 3: how you know about the 'local dominance of grasses/herbs throughout the Holocene'. Alkane distribution patterns or isotopes?

2, 10: 'valuable biomarkers' - for whom? A clear statement on the power of immortal molecules and the informative value could certainly improve the readability. Please check your manuscript throughout.

2, 12: 'increasingly used' – there a several groundbreaking studies that have already changed our understanding of the environment. Also, are there other biomarkers that can be used to trace primary productivity?

3, 1: what about ultra-small graphitization lines. Same same but different, other labs use conventional sample treatment at similar precision (« 10 $\mu$g C)

3, 2: how does MICADAS enable direct 14C dating of specific OC compounds? Do you mean online EA-AMS?

3, 20: 'this petrogenic contribution should lead to increased. . .'. In 14C, petrogenic is 14C depleted and thus it must be a matter of fraction size. Can you write this more clearly?

3, 22: what about microbial processing and impact, is it solely petrogenic?

9, 1-16: In my opinion, you miss the opportunity to inform the general audience about the principle of your measurement: you always measure a mean/median age of your individual or compound-class 14C n-alkanes because of the variable spatial origin and trajectories. This is central to understand that you integrate on spatial and temporal scale. Along these lines, is it correct to use these values for calibration absolute dating with IntCal14 (atmospheric 14C concentration) or better use 14C years only? Given you can, is your 14C age distribution a bell curve and how do you propagate the ana-

lytical uncertainties with the correction for petrogenic and the age dating?

9, 21: here you assume that your factor remains constant over the entire sequence while you source contributions likely are variable. Please add a statement.

10, 5-6: is it only erosion? What about sub-surface flow and export in addition to erosion of soil mineral horizon? Depending on the level of water saturation, would this impact your trajectories?

10, 27: are you sure your leaf wax n-alkanes are in-situ rather than originate from litter fall from vegetation on-site or transported by wind and water?

11, 5ff: Any thoughts on the role and extent of overprint by contemporary biological activity?

Your results point towards some spatial and time integrated value that is characteristic for a catchment. But how well does the sequence (depth profile) record the catchment changes in the past versus the soil development by contemporary vegetation?

11, 22: do you mean reworking?

11, 24: 'indicate high grass/herb percentages'. Please be specific. If you know the percentages, share it with the reader.

11, 25: 'not biased by pre-aging and reworking effects' – what do you mean? Please consider rephrasing

12, 3: 'this is caused' seems a quite strong statement. Please adjust

12, 4: 'this is further. . .' Please check that sentence carefully, it reads bulky.

12, 13: by anthropogenic activity: how? By any disturbance events, eg. deforestation?

12, 15: no older ages were determined? ÂňÂňÂňSo, this is the oldest?

12, 26: 'deposition than before'? please check

13, 7: only shale or also microbial?
* * *

---

## Referee Comment (RC2) · Anonymous Referee #2 · 14 Sep 2019

The authors of the manuscript investigate the age of sedimentary n-alkanes recovered from fluvial and paleosol deposits exposed at an outcrop in the upper Alazani valley in the Caucasus region. The manuscript describes analytical (i.e. pre-heating of the n-alkane fraction before 14C-dating) and procedural (estimation and correction for the contribution of petrogenic n-alkanes) improvements that will certainly be of interest to paleoclimatologists and biogeochemists using terrestrial biomarkers and their isotopic compositions in paleoclimate and paleoecology studies. It is a well-written manuscript supported by very detailed and finely executed figures. It fits the scope of this journal

and should be considered for publication provided the authors address several comments provided below.

MAJOR COMMENTS

FIRST, the contribution of n-alkanes from microbial sources to fluvial and paleosol deposits Throughout the manuscript the authors make a careful distinction between petrogenic n-alkanes that derive from organic-rich sedimentary rocks (14C dead, Jurassic black shales, in this study) and other n-alkanes from fluvial and paleosol sediments. The latter group are referred to as "leaf wax n-alkanes". While it is true that a major (and perhaps the largest) fraction in this group comprises leaf wax derived structures, it is quite likely that the group also contains microbial derived n-alkanes generated during pedogenic processes. Both molecular and isotopic composition of the "leaf wax" group can potentially be affected by the microbial source, e.g. Li et al. (2018, Org. Geochem. v. 115, 24-31), Wu et al. (2019, Org. Geochem., v. 128, 1-15). The authors, however, never mention this potential microbial source of n-alkanes. I suggest adding a discussion as to why this source is not considered to be important in general, and particularly when correcting F14C results for mass-dependent fractionation using 13C isotopes and when interpreting the results in section '3.5 Implications for leaf wax n-alkane-based paleoenvironmental reconstructions from our FSPS'.

SECOND, the level of detail when describing the study site The amount of detail given on pp. 3-4 when discussing the study site (section 2.1 Studied Site) and its geomorphological features is too excessive for the purposes of this manuscript. I suggest reducing it to a short paragraph and perhaps combining it with section 2.2 Stratigraphy.

MINOR COMMENTS

p. 1, line 33: "in-situ produced leaf wax n-alkanes" The use of the word in-situ is somewhat confusing here. Leaf wax n-alkanes can hardly be called in-situ when referring to soils and/or sediments. The term would probably fit more those n-alkanes that were produced within the soil (see above) during pedogenic processes.

p. 7, line 8: "All obtained 14C-ages are found in Table 1." Instead of this one-liner, it would be useful to have a short paragraph reminding the reader about the main goals of this paper and how the results obtained here can help with achieving these goals.

---

## Author Comment (AC1) · 3 Oct 2019

**Reply to Ulrich Hanke**

Dear Ulrich Hanke,

We thank you very much for taking the time to review our manuscript, for your valuable comments/suggestions and the points of discussion you raised. We will wisely revise our manuscript according to your suggestions. Please find our detailed responses below:

*The authors employ compound-class 14C measurements of long-chain leaf waxes (C_ 25) after removing the shorter chain lengths (C < 25) during the laboratory analyses to reduce the impact of petrogenic C (and reduce uncertainties from microbial reworking). However, even long chain leaf waxes contain a fraction of petrogenic carbon for which the authors introduce a factor and correct their 14C values to then discuss the pre-aging. Further, they translate their extracted 14C values to obtain calendar years yet hardly discuss the additional sensitivities introduced during "calibrating" with the 14C reference Intcal13. Or do the authors speak about 14C years? The former can be tricky because the paleo-fluvial sedimentary sequence seems to have features of recent biological activity and contemporary carbon that can complicate any attempt of absolute age dating.*

➔ Indeed, our heated and corrected 14C ages are reported as absolute/calibrated calendar age ranges because we wanted to test the chronostratigraphic integrity of the leaf wax n-alkanes in our fluvial sequence to assess their value for paleoenvironmental interpretations. To do so, we compared the n-alkane [14]C ages with [14]C-dated charcoal pieces from the sequence, the latter given as calibrated calendar ages and indicating the timing of sediment deposition. As a result, it appeared that n-alkane [14]C ages from the intensively developed paleosols fall into the timing of soil formation and are therefore chronostratigraphically consistent. In contrast, n-alkanes from fluvial sediment layers are much older than the timing of sediment deposition and are therefore chronostratigraphically not consistent. Since the latter n-alkanes are affected by pre-aging effects and furthermore represent a mixed signal of leaf waxes from the catchment and to a much lesser degree also from microbial activity, the calibrated and uncalibrated ages/age ranges give a very heterogenic integrated age information. However, we will give a more detailed description about our [14]C dating and calibration approach in the methods section.

*The excellent combination of molecular-level geochemical tools to trace the fate of carbon in past fluvial deposits is of great interest to earth scientists from various disciplines and, to my understanding, well suited for publication in HESS. The manuscript is scientific sound, entails adequate illustrations and details. However, the current version could benefit from (i) improve clarity in several sections (sentence length and perhaps language);*

➔ We will carefully revise the manuscript and check it for clarity and language.

*(ii) some clear statement on catchment-wide molecular-level 14C data that consists of an age distribution of 14C rather than a single value;*

➔ Will be done

*(iii) a statement on the informative value on ACL vs isotopes (13C and 14C);*

➔ Will be done in the Introduction

*(iv) discussion of data in the light of contemporary and pre-aged carbon;*

➔ We do not understand this comment, since we already carefully discussed the difference between pre-aged and contemporary n-alkanes.

*(v) some more informative details on potentials and limits on the geochemical constraints of molecular environmental 14C data compared to conventional 14C dating (in archeology).*

➔ Will be included into the introduction chapter.

*Probably some of my comments may already be included in the manuscript and might become clear after some careful checking/shortening of sentences and the addition of some clear statements. Overall a great study.*

➔ Thank you very much for this valuable and positive feedback!

*Some specific comments:*
*1,17: is 'direct' the correct term since you clean your fractions prior to 14C analyses?*

➔ "direct" means that we can directly [14]C-date the n-alkanes, and therefore derive an n-alkane-derived [14]C-age. Cleaning the n-alkane fraction aims to obtain a more homogeneous fraction. However, we can remove the term "direct".

*1,19-20: 'in-situ signal from local biomass': do you mean contemporary or on-site from litter fall?*
➔ We mean local biomass that derives on-site (in-situ) at the studied site, and that gets incorporated into the soil with the litterfall. However, as suggested by both reviewers, we will change the term in-situ to on-site.

*1, 30: 'were estimated': how do you determine the petrogenic C contributions? If only estimated, you may need to add some more details on the rationale and the precision of your approach*
➔ Petrogenic contributions are based on the quantified n-alkane proportion of Jurassic black shales from the catchment. We will specify this.

*2, 3: how you know about the 'local dominance of grasses/herbs throughout the Holocene'. Alkane distribution patterns or isotopes?*
➔ The vegetation distribution is derived from the n-alkane distribution pattern previously published in Bliedtner et al. 2018 (Quaternary Science Reviews 196, 62-79). We will specify this.

*2, 10: 'valuable biomarkers' - for whom? A clear statement on the power of immortal molecules and the informative value could certainly improve the readability. Please check your manuscript throughout.*
➔ Valuable biomarkers in paleoenvironmental research. Will be specified.

*2, 12: 'increasingly used' – there a several groundbreaking studies that have already changed our understanding of the environment. Also, are there other biomarkers that can be used to trace primary productivity?*
➔ Yes, you are right and we might rephrase this section. Of course there are other interesting biomarkers (e.g. fatty acids, sterols, bile acids etc.) that can be used to reconstruct climate, vegetation, soil erosion and/or human activity, but so far, leaf wax n-alkanes are the most prominent and most often used biomarker proxies.

*3, 1: what about ultra-small graphitization lines. Same but different, other labs use conventional sample treatment at similar precision (« 10 _g C)*
➔ Indeed, same labs use graphitization lines to date ultra-small samples, but it seems that sample preparation and measurement is less time and cost intensive when directly measured as $CO_2$ with a gas ion source.

*3, 2: how does MICADAS enable direct 14C dating of specific OC compounds? Do you mean online EA-AMS?*
➔ It is because the MICADAS enables [14]C dating of very small amounts of carbon (~20 μg C), although small amounts of carbon can also be analyzed by graphitization lines. The possibility to date those small amounts of carbon makes it possible to date specific OC compounds, such as leaf wax n-alkanes, that often occur only in smaller concentrations in sediment archives. It doesn't really matter if you use online EA-AMS or sealed tube combustion, although the analytical procedure with online EA-AMS is easier and less cost/time consuming. We will specify this section.

*3, 20: 'this petrogenic contribution should lead to increased: : :'. In 14C, petrogenic is 14C depleted and thus it must be a matter of fraction size. Can you write this more clearly?*
➔ You are right, and we will rewrite this section.

*3, 22: what about microbial processing and impact, is it solely petrogenic?*
➔ Indeed, we missed to state that n-alkanes can also originate from microbial sources or from microbial utilization. We will include such a statement into the introduction and a more detailed discussion into the discussion part. However, although we cannot completely rule out the influence of microbial utilization, we suggest that non-leaf wax-derived n-alkane contributions in our fluvial sequence are mostly of petrogenic origin. This is based on the fact that short- and mid-chain n-alkanes do not show an odd-over-even predominance, but in case of a

dominance of microbial processes we would expect an odd-over-even predominance in these chain lengths.

*9, 1-16: In my opinion, you miss the opportunity to inform the general audience about the principle of your measurement: you always measure a mean/median age of your individual or compound-class 14C n-alkanes because of the variable spatial origin and trajectories. This is central to understand that you integrate on spatial and temporal scale. Along these lines, is it correct to use these values for calibration absolute dating with IntCal14 (atmospheric 14C concentration) or better use 14C years only? Given you can, is your 14C age distribution a bell curve and how do you propagate the analytical uncertainties with the correction for petrogenic and the age dating?*

➜ We will include a more detailed description of our dating approach into the methods and discussion sections. You are right that we measured a mean $^{14}$C-age that can integrate over different spatial and temporal scales, what holds especially true for the n-alkanes derived from fluvial sediment layers. However, our main aim was to test the chronostratigraphic integrity of our n-alkanes in the fluvial sequence by comparing them with an independent charcoal chronology that gives the timing of sedimentation. By dating the n-alkanes from the fluvial sediment layers, we found that these are older than the sedimentation ages and therefore stratigraphically not consistent. Thus, they must be affected by pre-aging effects and contain a mixed leaf wax signal. For a better comparison between our different types of $^{14}$C-ages (from n-alkanes and charcoals), calibrated $^{14}$C ages seem to be best suited. Unfortunately, we have to note that error propagation after petrogenic correction is difficult and holds a certain uncertainty. Therefore, we simply used the measured $^{14}$C errors of the petrogenic-corrected $^{14}$C-ages. When corrected, the errors of the $^{14}$C-ages should basically become smaller and fall within the measured $^{14}$C-error. Thus, when using the errors mentioned above we will slightly overestimate the "true" $^{14}$C-error.

*9, 21: here you assume that your factor remains constant over the entire sequence while you source contributions likely are variable. Please add a statement.*

➜ Will be added.

*10, 5-6: is it only erosion? What about sub-surface flow and export in addition to erosion of soil mineral horizon? Depending on the level of water saturation, would this impact your trajectories?*

➜ We think that the degree of pre-ageing of the leaf wax n-alkanes from the catchment is mainly controlled by surficial soil erosion. It is true that organic material from catchment soils might also be relocated by sub-surface flow, but those effects are hard to quantify. Nevertheless, our $^{14}$C-results indicate that the catchment-derived leaf waxes carry an old and pre-aged signal from the catchment, most likely due to surficial soil erosion processes in the catchment that were also observed in the field.

*10, 27: are you sure your leaf wax n-alkanes are in-situ rather than originate from litter fall from vegetation on-site or transported by wind and water?*

➜ "In-situ" means that the leaf wax n-alkanes from the paleosols originate from local biomass/vegetation at the investigated site, i.e. it basically means "on-site". However, we cannot exclude that some of the n-alkanes in the paleosols were also transported by wind or water from neighbouring sites. However, the n-alkane-$^{14}$C-ages from the paleosols indicate they also that material could not have had a significant age-offset with the timing of sedimentation.

*11, 5ff: Any thoughts on the role and extent of overprint by contemporary biological activity? Your results point towards some spatial and time integrated value that is characteristic for a catchment. But how well does the sequence (depth profile) record the catchment changes in the past versus the soil development by contemporary vegetation?*

➜ In our previous publication (Bliedtner et al. 2018, Quaternary Science Reviews 196, 62-79) we report that based on micromorphological analyses we found some rhizo-microbial activity in the fluvial sediment layers. This indicates some overprinting by post-sedimentary processes. However, generally high carbonate contents in those layers indicate only short-time pedogenesis on the active floodplain what leads us to the assumption that post-sedimentary processes did not play an important role in our sequence. Moreover, the leaf wax $^{14}$C-results indicate that our sequence is at least partly well suited to record former vegetation changes: While the catchment signal is challenging to interpret in terms of paleoenvironmental changes due to a

spatially and temporarily integrated signal, the signal in the intensively developed paleosols originating from the investigated site is chronostratigraphically consistent and can therefore well be interpreted.

*11, 22: do you mean reworking?*
➔ Here we discuss the leaf wax signal from the (paleo)soils that is mostly incorporated from the local vegetation during soil development.

*11, 24: 'indicate high grass/herb percentages'. Please be specific. If you know the percentages, share it with the reader.*
➔ The percentages of grasses/herbs are only semi-quantitative estimates that are based on a correction approach that is discussed more in detail in Bliedtner et al. 2018 (QSR 196, 62-79). Given that the percentages can exceed 100% a statement about precise percentages would therefore be misleading. However, we will provide a more detailed description of the grass/herb percentage ratio here.

*11, 25: 'not biased by pre-aging and reworking effects' – what do you mean? Please consider rephrasing*
➔ We mean that leaf waxes from the intensively developed (paleo)soils are chronostratigraphically consistent and not affected by pre-aging and reworking. We will rephrase this section accordingly.

*12, 3: 'this is caused' seems a quite strong statement. Please adjust*
➔ Will be adjusted.

*12, 4: 'this is further: : :' Please check that sentence carefully, it reads bulky.*
➔ Will be changed.

*12, 13: by anthropogenic activity: how? By any disturbance events, eg. deforestation?*
➔ Mainly by deforestation and land-use in the catchment. Will be specified.

*12, 15: no older ages were determined? ¡n¡n¡nSo, this is the oldest?*
➔ Yes, this is the oldest age. However, we were not able to date the two lowermost fluvial sediment layers and therefore do not have age information of them.

*12, 26: 'deposition than before'? please check*
➔ Will be changed.

*13, 7: only shale or also microbial?*
➔ Mainly shale, but we will include a more detailed discussion about microbial influences in the discussion part.

---

## Author Comment (AC2) · 3 Oct 2019

**Reply to Referee#2**

Dear Referee#2,

We thank you for taking the time to review our manuscript and for your valuable comments/suggestions. We will carefully revise our manuscript according to your suggestions. Please find our detailed responses below:

*MAJOR COMMENTS*
*FIRST, the contribution of n-alkanes from microbial sources to fluvial and paleosol deposits: Throughout the manuscript the authors make a careful distinction between petrogenic n-alkanes that derive from organic-rich sedimentary rocks (14C dead, Jurassic black shales, in this study) and other n-alkanes from fluvial and paleosol sediments. The latter group are referred to as "leaf wax n-alkanes". While it is true that a major (and perhaps the largest) fraction in this group comprises leaf wax derived structures, it is quite likely that the group also contains microbial derived n-alkanes generated during pedogenic processes. Both molecular and isotopic composition of the "leaf wax" group can potentially be affected by the microbial source, e.g. Li et al. (2018, Org. Geochem. v. 115, 24-31), Wu et al. (2019, Org. Geochem., v. 128, 1-15). The authors, however, never mention this potential microbial source of n-alkanes. I suggest adding a discussion as to why this source is not considered to be important in general, and particularly when correcting F14C results for mass-dependent fractionation using 13C isotopes and when interpreting the results in section '3.5 Implications for leaf wax n-alkane-based paleoenvironmental reconstructions from our FSPS'.*

➔ Indeed, we missed to state in our manuscript that n-alkanes can also originate from microbial sources or microbial utilization, and thus can produce much younger $^{14}$C ages. Especially the potential buildup of long-chain n-alkanes by such organisms as described by Li et al. (2018) can be a serious issue that might complicate $^{14}$C dating and the paleoenvironmental interpretation of the respective leaf wax proxies. Therefore, we will include such a statement into the introduction as well as a more detailed discussion into the discussion part as suggested by the reviewer. However, although we cannot completely rule out the influence of microbial utilization, we suggest that non-leaf wax-derived n-alkane contributions in our fluvial sequence are mostly of petrogenic origin. This is based on the fact that short- and mid-chain n-alkanes do not show an odd-over-even predominance, but in case of a dominance of microbial processes we would expect an odd-over-even predominance in these chain lengths.

*SECOND, the level of detail when describing the study site The amount of detail given on pp. 3-4 when discussing the study site (section 2.1 Studied Site) and its geomorphological features is too excessive for the purposes of this manuscript. I suggest reducing it to a short paragraph and perhaps combining it with section 2.2 Stratigraphy.*

➔ In this case, we do not agree with the reviewer, since we feel that our manuscript should be able to stand alone, i.e. without the need to read our formerly published results. Therefore, to allow the readers to easily evaluate the geomorphic situation of the study site we suggest to keep the description as it is.

*MINOR COMMENTS*
*p. 1, line 33: "in-situ produced leaf wax n-alkanes" The use of the word in-situ is somewhat confusing here. Leaf wax n-alkanes can hardly be called in-situ when referring to soils and/or sediments. The term would probably fit more those n-alkanes that were produced within the soil (see above) during pedogenic processes.*

➔ We will change the term in-situ to on-site.

*p. 7, line 8: "All obtained 14C-ages are found in Table 1." Instead of this one-liner, it would be useful to have a short paragraph reminding the reader about the main goals of this paper and how the results obtained here can help with achieving these goals.*

➔ Will be done

---

## Author Response (AR3)

Dear Laurent Pfister,

we would like to thank you again as the handling editor for your effort with our manuscript and we thank the reviewer for the supportive review and the recommendations. We have replied to the reviewer suggestions in the point-to-point reply below and have revised our manuscript accordingly. Please find also below the revised manuscript with marked-up changes.

With best wishes on behalf of the co-authors,
Marcel Bliedtner

**Reply to the Referee**

*1-17: "...derives..." sure? Expected: yields, results, provides* → Will be changed to yields.

*1-19: "...store..."? 'integrate' would link the following two sections better* → Will be changed to integrate.

*11-4: please change "5" to 'five' to support readability* → Will be changed.

*11-14&21: No. or # instead of Nr.* → Will be changed.

*11-16: "...with solvent evaporated..." please check* → We removed "with solvent evaporated".

*13-10: suggest adding 'here' after "that _ ..." as this refers to your site* → will be changed.

*13-19: "...closer to it's true value..." of what you expected, anticipated? please specify* → We expect that the heated and corrected $^{14}$C-ages should be leaf wax-derived $^{14}$C-ages. We will therefore delete the term "true" and specify by "leaf wax-derived $^{14}$C-ages".

*16-13: "obviously" - you used this several times throughout yet I wondered whether 'apparently, seemingly, etc. could fit the purpose better* → We will delete the term "obviously" throughout the manuscript or change it to apparently.

*18-21: instead of "only gave minimum values", consider "provide an (conservative) estimate" for the age offsets.* → Will be changed.

*19-4: While you changed throughout the manuscript, still use the word "directly" for this multi-step procedure in the conclusions. overlooked?* → Will be changed.

*19-13: again, please add some descriptive words to "true" to reduce confusion for the general audience* → Will be changed.

*19-29: "Similar..." can you add references or perhaps simply rephrase? This current wording reads incomplete.* → *We prefer to not include references in the conclusions, but will rephrase the sentence.*

*20-3: how about 'provides an avenue...' instead of is an important step...* → We will change to "is an important and valuable tool".

[revised manuscript text omitted]